# Tree Structure for the Categorical Wasserstein Weisfeiler-Lehman Graph Kernel

**Keishi Sando**                                                                    *sando.keishi.sp@alumni.tsukuba.ac.jp*
*Department of Statistical Science*
*The Graduate University for Advanced Studies*

**Tam Le**                                                                                                 *tam@ism.ac.jp*
*The Institute of Statistical Mathematics*

**Hideitsu Hino**                                                                                        *hino@ism.ac.jp*
*The Institute of Statistical Mathematics / Waseda University*

**Reviewed on OpenReview:** *https://openreview.net/forum?id=VwoSsFK22P*

## Abstract

The Wasserstein Weisfeiler-Lehman (WWL) graph kernel is a popular and efficient approach, utilized in various kernel-dependent machine learning frameworks for practical applications with graph data. It incorporates optimal transport geometry into the Weisfeiler-Lehman graph kernel, to mitigate the information loss inherent in aggregation strategies of graph kernels. While the WWL graph kernel demonstrates superior performance in many applications, it suffers a drawback in its computational complexity, i.e., at least $\mathcal{O}(n_1 n_2)$, where $n_1, n_2$ denote the number of vertices in the input graphs. Consequently, it hinders the practical applicability of the WWL graph kernel, especially in large-scale settings. In this paper, we propose the *Tree Wasserstein Weisfeiler-Lehman* (TWWL) algorithm, which leverages a *tree structure* to scale up the exact computation of the WWL graph kernel for graph data with categorical node labels. In particular, the computational complexity of the TWWL algorithm is $\mathcal{O}(n_1 + n_2)$, which enables its application to large-scale graphs. Numerical experiments demonstrate that the performance of the proposed algorithm compares favorably with baseline kernels, while its computation is several orders of magnitude faster than the classic WWL graph kernel. This paves the way for applications in large-scale datasets where the WWL kernel is computationally prohibitive.

## 1 Introduction

Graph data are a popular structure and play a fundamental role in many applications (Barabási & Oltvai, 2004; Pržulj, 2007; Barabási & Pósfai, 2016; Petric Maretic et al., 2019; Xu et al., 2019; Nguyen et al., 2021; Chen et al., 2022; Vincent-Cuaz et al., 2022; Le et al., 2022; Kong et al., 2023). To incorporate such structured data into machine learning frameworks, several graph kernels (Vishwanathan et al., 2010; Borgwardt et al., 2020; Kriege et al., 2020; Nikolentzos et al., 2021) have been proposed in the literature, which measure similarity of input graphs by aggregating their local features, e.g., histogram kernels (Nikolentzos et al., 2021) compare multisets counting on nodes or edges, path-based kernels (Borgwardt & Kriegel, 2005; Feragen et al., 2013) accumulate similarities along shortest paths, and graphlet kernels (Pržulj, 2007; Shervashidze et al., 2009) evaluate the frequency of small subgraph structures. Despite their effectiveness, these kernels face certain significant challenges. A primary limitation is poor scalability with the number of nodes in input graphs. For examples, the shortest path kernel relies on the pairwise shortest paths between all graph nodes, which typically requires $\mathcal{O}(n^3)$, where $n$ is the number of nodes in a graph. The $k$-node graphlet kernel requires about $\mathcal{O}(n^k)$ induced subgraphs, forcing users either to perform an exhaustive count or to approximate it by sampling; both approaches become prohibitive even for moderate $k$.

To overcome this challenge and to enhance the expressiveness of graph kernels, Shervashidze et al. (2011) proposed the Weisfeiler-Lehman (WL) kernels. By coupling iterative neighborhood aggregation with relabeling, the WL kernel achieves neighborhood-aware representations while retaining computational efficiency. Moreover, Togninalli et al. (2019) leveraged optimal transport (Villani, 2009) to enrich the representational power of the WL kernel, and proposed the Wasserstein Weisfeiler-Lehman (WWL) kernel. Notice that several traditional graph kernels, e.g., graphlet kernels, shortest-path kernels, and WL subtree kernel (Shervashidze et al., 2011), typically enumerate substructures and then aggregate them by frequency counting or averaging. Such a pooling approach collapses the empirical distribution of substructures to its first-order statistics, eliminating information about how different patterns co-occur or how their attributes are distributed. The WWL kernel overcomes this information loss by comparing the entire distributions of node embeddings via the Wasserstein distance (Peyré & Cuturi, 2020), thus retaining intricate structural information.

Although the WWL graph kernel strengthens the ability to capture complex graph structures, the Wasserstein distance within its computation process raises concerns about its scalability (i.e., supercubic computational complexity). Since this challenge stems from the inherent difficulty in optimal transport theory, various acceleration strategies have been explored in many domains. There are two main acceleration strategies for the computation of the Wasserstein distance: (i) the sliced Wasserstein distance (SWD) (Bonneel et al., 2015), which relies on one-dimensional projections of the supports of input measures, and exploits the closed-form expression of univariate optimal transport; and (ii) an entropic regularization approach (Cuturi, 2013), which can be solved efficiently by the celebrated Sinkhorn algorithm (Sinkhorn, 1964). Extensive research has been conducted related to these approaches. Perez et al. (2024) proposed to utilize the SW approach to *approximate* the WWL kernel, to scale up its computation, which is applicable to large-scale datasets with continuous node attributes. In the Sinkhorn algorithm, various proposals have been introduced in the literature to further reduce its computational cost, including a greedy strategy (Altschuler et al., 2017), kernel matrix sparsification (Li et al., 2023), and low-rank factorization of the coupling (Scetbon et al., 2021). Additionally, Tang et al. (2024) proposed to consider a Newton-type method with sparse Hessian matrix to accelerate its convergence. In a complementary direction to SWD and Sinkhorn, a multiscale approach (Mérigot, 2011) solves the optimal transport problem on a coarse-to-fine hierarchy with warm starts, which results in reduced runtimes. Another line of work develops a proximal point method for computing the exact Wasserstein distance (Xie et al., 2019). Using a Bregman proximal point scheme with inexact Sinkhorn projections, this algorithm still converges to the exact solution, alleviating entropic bias.

In contrast to the above strategies, by restricting the space to a tree structure, it has been shown that the Wasserstein distance can be expressed in closed form (Indyk & Thaper, 2003; Le et al., 2019b). This variant is commonly referred to as the Tree Wasserstein distance (TWD). The computational efficiency of the TWD has been leveraged in various applications. Le et al. (2019b) considered the TWD as a generalization of the univariate Wasserstein distance, and then proposed the tree-sliced Wasserstein distance as a variant of the SWD. As the choice of a tree structure plays a crucial role in the TWD, Yamada et al. (2022) proposed an algorithmic approach for estimating the edge weights to approximate the Wasserstein distance. Lin et al. (2025) addressed the issue of how to construct a tree by embedding features into a multiscale hyperbolic space via diffusion geometry, decoding the latent feature hierarchy into an explicit tree, and defining the TWD on the learned tree. Acceleration algorithms leveraging the TWD have also been proposed for the Wasserstein barycenter problem (Le et al., 2019a; Takezawa et al., 2022) and for metric learning using Wasserstein singular vectors (Düsterwald et al., 2025).

Following the original formulation of the WWL kernel, subsequent work has advanced both its computational efficiency and its performance. On the performance front, Titouan et al. (2019) proposed the Fused Gromov-Wasserstein distance, which jointly aligns node attributes and graph structure information within a single optimal transport objective. Chen et al. (2022) introduced the Wasserstein-Lehman distance, a polynomial-time optimal transport metric between labeled Markov chains, that lifts WL iterations to a hierarchy of probability measures. This yields stronger discriminative power than WWL and provides a practical WL lower bound for tractability. Schulz et al. (2022) proposed a relaxed Weisfeiler-Lehman subtree kernel that encodes WL labels as rooted trees and compares them via tree edit metric. This relaxes the hard 0/1 equality into flexible similarity. Chuang & Jegelka (2022) construct, for each graph, the multiset of computation trees rooted at every vertex and, by defining hierarchical optimal transport between trees, introduced an optimal

transport distance between these multisets as a graph metric. With respect to computational scalability, Perez et al. (2024) proposed an approximation method that accelerates the WWL kernel for graphs with continuous node attributes, and generic accelerations for Wasserstein computation are likewise applicable.

A challenge in graph-based machine learning is the trade-off between the expressive power of graph comparison and their computational scalability. While advanced graph metrics can capture both global and local structure, they often demand substantial computational costs. Therefore, developing approaches that achieve an effective balance is essential for tackling the large-scale datasets in modern applications. In this paper, we address the acceleration of the WWL graph kernel for datasets with categorical node labels by utilizing the TWD. Specifically, we first show that the labels generated by the WL algorithm inherently exhibit a tree structure, and that the resulting path metric on the tree exactly preserves the Hamming distance between the labels. This results in a closed-form expression to calculate the WWL kernel for categorical node labels. One notable advantage of our approach over the Sinkhorn approach is its predictable runtime. Whereas the Sinkhorn solver requires a quadratic complexity over the number of iterations to converge, the computational cost of our approach is fixed once the WL algorithm is completed.

**Contribution.** Our contribution is three-fold:

1. We present that the relationship among generated labels by the WL algorithm forms a tree structure.

2. We show that the proposed algorithm, *Tree Wasserstein Weisfeiler-Lehman* algorithm, is an exact and accelerated method for computing the WWL graph kernel designed for graph datasets with categorical node labels.

3. Numerical experiments demonstrate that the computational complexity of the proposed algorithm outperforms the original WWL kernel algorithm without sacrificing its performances. Moreover, our algorithm makes it feasible to apply the WWL kernel to large-scale categorical-label graph datasets that are computationally prohibited for the original WWL graph kernel algorithm.

**Organization.** The paper is organized as follows: Section 2 is devoted to notations in this paper. In Section 3, we review the related works that are necessary to construct our algorithm. Our algorithm and its properties are described in Section 4. Section 5 presents the numerical experiments, and we conclude the paper in Section 6.

## 2 Notations

In this section, we introduce the notations used in subsequent sections. Let $\mathcal{G} = (V, E)$ be an undirected graph, where $V$ and $E$ denote the set of vertices and edges, respectively. We write the neighborhood of a vertex $v$ by $\mathcal{N}_v \subset V$. We denote a set of initial categorical labels by $\Sigma_0$, and say that a graph is *labeled* if each of its vertices is assigned a label from $\Sigma_0$.[1] Let $\boldsymbol{\mathcal{G}} := \{\mathcal{G}_1 = (V_1, E_1), \ldots, \mathcal{G}_N = (V_N, E_N)\}$ be a set of graphs, $N$ be the number of graphs, $n_i$ be the number of vertices in $\mathcal{G}_i$, $n := \sum_{i=1}^{N} n_i$ be the total number of vertices in $\boldsymbol{\mathcal{G}}$, and $\bar{n} := \frac{n}{N}$ be the average number of vertices per graph. Throughout this paper, we assume that all graphs in $\boldsymbol{\mathcal{G}}$ are labeled and share the same label set $\Sigma_0$. Furthermore, we write $\mathcal{V} := \bigcup_{i=1}^{N} V_i$. A function $l_0 : \mathcal{V} \to \Sigma_0$ is used to denote the initial label of vertices.

A sequence of vertices $(v_1, \ldots, v_k)$ in a graph $\mathcal{G}$ is called a *walk* if an edge exists between every pair of consecutive vertices. If all vertices in the walk are distinct, the walk is called a *path*. A sequence $(v_1, \ldots, v_k)$ is called a *cycle* if its first and last vertices coincide, the vertices $v_1, \ldots, v_{k-1}$ are distinct, and for every pair of consecutive vertices there exists an edge between them. A graph is *connected* if, for every pair of vertices there exists a path. We denote an edge weight function by $w : E \to \mathbb{R}_{\geq 0}$.

A connected graph without cycles is called a *tree*. We denote a tree by $\mathcal{T} = (V(\mathcal{T}), E(\mathcal{T}))$ rather than $\mathcal{G}$. The vertices and edge sets of $\mathcal{T}$ are denoted $V(\mathcal{T})$ and $E(\mathcal{T})$, respectively. Note that for any pair of vertices

---

[1]For graphs without node labels, We assign dummy labels to each vertex.

in a tree, there exists a unique path connecting them. This is because the existence of two distinct paths would necessarily imply the presence of a cycle, contradicting the definition of a tree.

We write $\mathcal{P}(u, v)$ as a set of edges that constitute the path between $u, v \in V(\mathcal{T})$. Given a tree $\mathcal{T}$ with a weight function $w$, the *path metric* between $u, v \in V(\mathcal{T})$ is defined as $d_{(\mathcal{T}, w)}(u, v) \coloneqq \sum_{e \in \mathcal{P}(u, v)} w_e$, where $w_e \coloneqq w(e)$. We denote a tree rooted at $r \in V(\mathcal{T})$ by $\mathcal{T}_r$, and a set of nodes in a subtree rooted at $u \in V(\mathcal{T}_r)$ by $\Gamma_u \coloneqq \{v \in V(\mathcal{T}_r) \mid u \in \mathcal{P}(v, r)\}$. For an edge $e \in E(\mathcal{T}_r)$, we write $u_e \in V(\mathcal{T}_r)$ for the node closer to the root and $v_e \in V(\mathcal{T}_r)$ for the node farther away from the root node.

Figure 1 illustrates the notations of path metric, subtree, and edge endpoints introduced above. The path metric between vertices $y$ and $z$ can be calculated as $d_{(\mathcal{T}_r, w)}(y, z) = w_1 + w_2 + w_3$, the subtree rooted at $y$ is indicated by the region outlined in blue, and $\Gamma_y$ denotes the set of vertices contained within the region. For the edge $e$ that joins vertices $x$ and $z$, the endpoint closer to the root is $u_e = x$, whereas the endpoint farther from the root is $v_e = z$.

# 3 Related work

Our algorithmic approach accelerates the categorical WWL graph kernel, which combines the WL algorithm with the Wasserstein distance, by leveraging the tree structure over labels within the WL procedure. In this section, we review the necessary background and related works that support our approach.

## 3.1 Wasserstein distance

The Wasserstein distance (Villani, 2009; Peyré & Cuturi, 2020) is a metric between probability measures supported on a given metric space. Let $(\Omega, d)$ be a metric space and $\mu, \nu$ be probability measures supported on $\Omega$. The $L^p$-Wasserstein distance for $p \geq 1$ is defined as

$$
W^p(\mu, \nu) \coloneqq \left( \inf_{\pi \in \Pi(\mu, \nu)} \int d(x, y)^p \pi(dx, dy) \right)^{1/p},
$$

where $\pi$ is a probability measure on $\Omega \times \Omega$ whose marginals are $\mu$ and $\nu$, called a *coupling*, and $\Pi(\mu, \nu)$ is the set of all couplings of $\mu$ and $\nu$. In this paper, we focus on the $L^1$-Wasserstein distance, which we denote by $W_d$, i.e., optimal transport with ground cost $d$.

## 3.2 Weisfeiler-Lehman algorithm

Shervashidze et al. (2011) proposed to incorporate the concept of the Weisfeiler-Lehman test (Weisfeiler & Leman, 1968) into graph kernels. The WL test is a procedure for graph isomorphism problem that iteratively refines vertex labels based on their neighborhood structure. This approach yields features that are both highly expressive, by capturing latent graph structure, and computationally efficient, thereby addressing the scalability issues that hindered various graph kernels.

The WL algorithm iteratively assigns a new label to each vertex, treating the resulting sequence of labels as the feature of the vertex. Let $H \in \mathbb{N}$ denote the maximum number of iterations, $\Sigma_h$ denote the set of all labels generated at the $h$-th iteration, and $\Sigma \coloneqq \bigcup_{h=0}^{H} \Sigma_h$ denote the set of all labels. At the $h$-th iteration in the WL algorithm, the new label for vertex $v$ is generated as follows, based on its previous label and the set of labels in its neighborhood:

$$
l_h(v) \coloneqq \text{hash}\left(l_{h-1}(v), \mathcal{N}_{v, h-1}\right), \tag{1}
$$

where we write $\mathcal{N}_{v, h-1} \coloneqq \{l_{h-1}(u) \mid u \in \mathcal{N}_v\}$ and handle $\mathcal{N}_{v, h-1}$ in its sorted form. The hash function (Cormen et al., 2009) is defined as a function commonly used in computer science that returns the same output for identical inputs and returns the distinct value for different inputs. A general description of hash functions is provided in Appendix A. Lemma 1 immediately follows from the definition of the hash function.

**Lemma 1.** *For every $\sigma \in \Sigma_h$, there exists a unique label sequence $(\sigma_0, \ldots, \sigma_{h-1})$, $\sigma_i \in \Sigma_i$ that generates $\sigma$.*

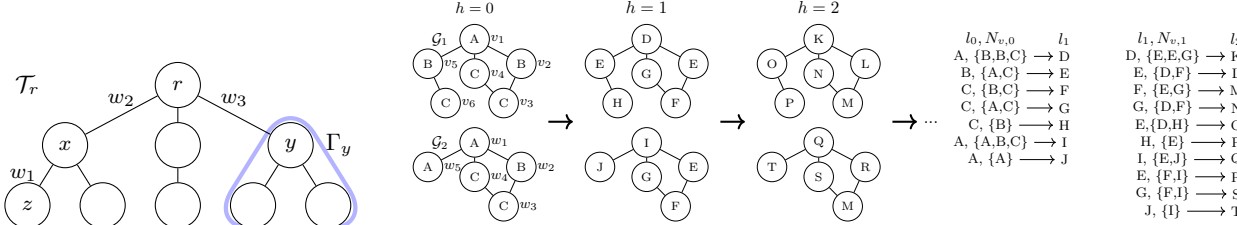

Figure 1: An example of tree notations.

Figure 2: An example of the WL algorithm for given graphs with $\Sigma_0 = \{A, B, C\}$.

Figure 2 illustrates the iterative process of the WL algorithm on labeled graphs $\boldsymbol{\mathcal{G}} = \{\mathcal{G}_1, \mathcal{G}_2\}$. In this figure, the new labels generated at each step are represented by Latin characters that are not used in the previous steps, reflecting the injectivity of the hash function. Vertices $v_2$ and $v_5$ start with identical labels and the same multiset of neighbor labels, meaning that they remain indistinguishable after the first iteration. However, as structural information from farther neighborhoods is propagated, their local contexts diverge. This divergence leads to them being assigned distinct labels at $h = 2$. This example demonstrates how the WL algorithm effectively propagates structural information to eventually differentiate vertices that initially appear identical from a local viewpoint.

### 3.3 Weisfeiler-Lehman subtree kernel

The WL subtree kernel, introduced by Shervashidze et al. (2011), is a member of a graph kernel family that leverages the WL algorithm. Based on the label set $\Sigma$ generated by the WL algorithm, let $c(\sigma, \mathcal{G})$ denotes the count of a label $\sigma$ in $\mathcal{G}$. Then, the WL subtree feature for $\mathcal{G}$ is defined as

$$\phi(\mathcal{G}) := \left(c(\sigma_1, \mathcal{G}), \ldots, c(\sigma_{|\Sigma|}, \mathcal{G})\right).$$

We use $\langle \cdot, \cdot \rangle$ to denote an inner product, then the WL subtree kernel on two graphs $\mathcal{G}_1, \mathcal{G}_2$ is defined as

$$k_{\text{WLsubtree}}(\mathcal{G}_1, \mathcal{G}_2) := \langle \phi(\mathcal{G}_1), \phi(\mathcal{G}_2) \rangle.$$

The WL subtree kernel is positive semidefinite and can be computed in $\mathcal{O}(Hm)$, where $m$ is the total number of edges in $\mathcal{G}_1$ and $\mathcal{G}_2$.

### 3.4 Wasserstein Weisfeiler-Lehman graph kernel

Most classical graph kernels aggregate substructures by a simple pooling approach, which reduces the empirical distribution to first-order statistics and discards information about pattern co-occurrence and attribute variation. To reflect this information in the comparison, Togninalli et al. (2019) proposed the WWL kernel. The key idea is to treat the multiset of WL node labels as an empirical distribution over the label space and to compare the similarity between two graphs via the Wasserstein distance on that space for the WWL kernel.

For labeled graphs, the ground distance between each pair of vertices $v \in V_i$, $w \in V_j$ is defined by the Hamming distance between their WL label sequences:

$$d_{\text{Ham}}(v, w) := \sum_{h=0}^{H} \rho\left(l_h(v), l_h(w)\right), \quad \text{where} \quad \rho(a, b) = \begin{cases} 1 & a \neq b, \\ 0 & \text{otherwise.} \end{cases} \tag{2}$$

In Togninalli et al. (2019), Equation 2 is normalized by $H + 1$. Since normalization by a constant does not affect the results, we omit this factor to simplify the discussion.

Due to the uniqueness of the label sequence that generates a label $\sigma \in \Sigma_H$ and the definition of the hash function, the Hamming distance $d_{\text{Ham}}$ can be regarded as a metric on $\Sigma_H$. Assigning a uniform probability

measure to each vertex induces empirical measures on that label space. Let $\mu, \nu$ be probability measures on $\Sigma_H$ for $\mathcal{G}_1$ and $\mathcal{G}_2$, respectively. The Wasserstein distance $W(\mathcal{G}_1, \mathcal{G}_2)$ is then computed by

$$W(\mathcal{G}_1, \mathcal{G}_2) := W_{d_{\mathrm{Ham}}}(\mu, \nu) = \inf_{\pi \in \Pi(\mu, \nu)} \int d_{\mathrm{Ham}}(x, y) \pi(dx, dy), \tag{3}$$

and the corresponding kernel is defined by

$$k_W(\mathcal{G}_1, \mathcal{G}_2) := e^{-\lambda W(\mathcal{G}_1, \mathcal{G}_2)}, \tag{4}$$

where $\lambda$ is a positive hyperparameter. Togninalli et al. (2019) employed either the network simplex algorithm (Orlin, 1997) to compute the Wasserstein distance or the Sinkhorn algorithm (Sinkhorn, 1964; Cuturi, 2013) to approximate it.

### 3.5 Tree Wasserstein distance

The Tree Wasserstein distance was initially introduced by Indyk & Thaper (2003), and implicitly utilized in *UniFrac* (Lozupone & Knight, 2005), which quantifies the similarity between two microbial communities on a reference phylogenetic tree, then was rigorously formalized by Evans & Matsen (2012). When the ground space is a weighted tree, the 1-order Wasserstein distance admits a closed form that can be evaluated in $\mathcal{O}(|E|)$. In this section, we define the Wasserstein distance on a tree and present prior work showing that it admits a closed-form expression.

Given that $d_{\mathcal{T}}$ is a path-length metric on $\mathcal{T}$ and $\mu, \nu$ are probability measures on $\mathcal{T}$, we refer to the $L^1$-Wasserstein distance on $\mathcal{T}$ as the Tree Wasserstein distance and define it below:

$$W_{d_{\mathcal{T}}}(\mu, \nu) := \inf_{\pi \in \Pi(\mu, \nu)} \int d_{\mathcal{T}}(x, y) \pi(dx, dy).$$

Prior work has established that the $L^1$-Wasserstein distance on a tree admits the following closed-form expression.

**Theorem 1** ((Evans & Matsen, 2012; Le et al., 2019b)). *The Tree Wasserstein distance can be written as follows:*

$$W_{d_{\mathcal{T}}}(\mu, \nu) = \sum_{e \in E(\mathcal{T})} w_e \left| \mu(\Gamma(v_e)) - \nu(\Gamma(v_e)) \right|.$$

Computing this expression requires only a single post-order traversal of the tree. When the tree degenerates to a path, the formula reduces to the standard univariate Wasserstein distance, highlighting TWD as a natural generalization. While not adopted in our setting, the closed-form representation in Theorem 1 can be written in matrix notation. For further details, we refer the reader to Takezawa et al. (2021).

## 4 Main Algorithm

We propose the *Tree Wasserstein Weisfeiler-Lehman* (TWWL) algorithm that enables us to calculate the Wasserstein distance based on the Hamming distance precisely and efficiently, without entropic regularization. In our algorithm, we focus on the relationship between $l_h(v)$ and $l_{h-1}(v)$ in Equation 1.

**Definition 1** (Parent-Child relationship among labels). For $\sigma \in \Sigma_h$ and $\tau \in \Sigma_{h+1}$, if there exists a label set $S \subset \Sigma_h$ such that $\tau = \mathrm{hash}(\sigma, S)$, then $\sigma$ is considered a parent label of $\tau$, or equivalently, $\tau$ is a child label of $\sigma$.

It is clear from the property of the hash function that the following holds.

**Lemma 2.** *Let $u, v$ be arbitrary vertices in $\mathcal{V}$. If there exists $h_0 < H$ such that $l_{h_0}(u) \neq l_{h_0}(v)$, then $\forall h \in \{h_0, \ldots, H\}$, $l_h(u) \neq l_h(v)$ holds.*

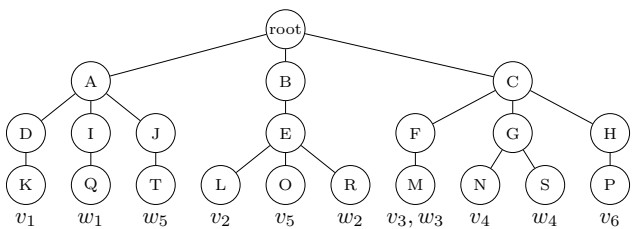

Figure 3: An illustration of a label tree based on graphs in Figure 2.

The TWWL algorithm yields a tree structure for labels, called the *Label Tree*, and applies the Tree Wasserstein distance. The label tree consists of nodes corresponding to the labels produced by the WL algorithm and preserves the relation in Definition 1. The steps for constructing a label tree are described in Algorithm 1. The label tree is constructed to satisfy the following properties: (i) except for the root node, each node is in one-to-one correspondence with a label in $\Sigma$; (ii) for every label in $\Sigma_0$, there is an edge connecting it to the root node; (iii) an edge exists between any two labels that satisfy the relation defined in Definition 1; (iv) each edge is assigned a weight of $1/2$.

Figure 3 illustrates the label tree constructed from the example in Figure 2. Nodes at depth $d$ in the label tree correspond to the labels generated at the $(d-1)$-th iteration of the WL algorithm. Taking the label tree properties into consideration, one can intuitively verify that the Hamming distance between $v_1$ and $w_1$, as defined in Equation 2 with $H = 2$, coincides with the length of the shortest path on the tree. The following two lemmas provide a formal justification for these claims.

**Lemma 3.** *A label tree obtained by Algorithm 1 is a tree.*

*Proof.* We begin by proving the connectivity of the label tree, i.e., that there exists a walk between any $\sigma, \tau \in \Sigma$. By Lemma 1 and the property (iii) of the label tree, there exist label sequences $(\sigma_0, \ldots, \sigma)$, $(\tau_0, \ldots, \tau)$ that form a path in the label tree. The property (ii) implies that there exist edges connecting the root to $\sigma_0$ and to $\tau_0$. Therefore, a walk exists between $\sigma$ and $\tau$.

Next, we show that the label tree contains no cycle. Suppose, for the sake of contradiction, that there exists a cycle $(\sigma, \tau_1, \ldots, \tau_k, \sigma)$ in the label tree for some label $\sigma \in \Sigma$. Without loss of generality, assume that $\sigma$ is the deepest vertex in the cycle. Then, both $\tau_1$ and $\tau_k$ must be parent nodes of $\sigma$. By a property of the hash function, $\tau_1 = \tau_k$ holds. However, this equality contradicts the definition of a cycle, which requires that the intermediate vertices are distinct. Therefore, no cycle exists in the label tree. $\qquad\square$

**Lemma 4.** *The mapping $\phi : \Sigma_H \to \mathcal{T}$ obtained by Algorithm 1 is distance-preserving.*

*Proof.* We denote $\phi(\sigma)$ simply as $\sigma$. We show that for any $\sigma_H, \tau_H \in \Sigma_H$, $d_{\mathrm{Ham}}(\sigma_H, \tau_H) = d_{(\mathcal{T},1/2)}(\sigma_H, \tau_H)$ holds. Lemma 1 implies that there exist label sequences $(\sigma_0, \ldots, \sigma_H)$, $(\tau_0, \ldots, \tau_H)$ that satisfy the parent-child relationship. From Lemma 2, we consider the following three cases:

Case 1: $\forall h \in \{0, \ldots, H\}$, $\sigma_h = \tau_h$. Then, we have both $d_{\mathrm{Ham}}(\sigma_H, \tau_H)$ and $d_{(\mathcal{T},1/2)}(\sigma_H, \tau_H)$ are 0.

Case 2: $\sigma_0 \neq \tau_0$. Then, it follows from Lemma 2 that $d_{\mathrm{Ham}}(\sigma_H, \tau_H) = H + 1$. Additionally, Lemma 3 implies that the sequence $(\sigma_H, \ldots, \sigma_0, \mathrm{root}, \tau_0, \ldots, \tau_H)$ is the unique path between $\sigma_H$ and $\tau_H$. Thus, $d_{(\mathcal{T},1/2)}(\sigma_H, \tau_H) = H + 1$.

Case 3: $\exists h_0 \in \{1, \ldots, H\}$, $\sigma_{h_0-1} = \tau_{h_0-1}$ and $\sigma_{h_0} \neq \tau_{h_0}$. Then, by Lemma 2, $d_{\mathrm{Ham}}(\sigma_H, \tau_H) = H + 1 - h_0$ is satisfied and the sequence $(\sigma_H, \ldots, \sigma_{h_0}, \sigma_{h_0-1} = \tau_{h_0-1}, \tau_{h_0}, \ldots, \tau_H)$ is a unique path between $\sigma_H$ and $\tau_H$. This implies $d_{(\mathcal{T},1/2)}(\sigma_H, \tau_H) = H + 1 - h_0$.

Therefore, Lemma 4 holds for all cases. $\qquad\square$

The edge weight $1/2$ in property (iv) ensures that the path-length metric on the label tree coincides with the Hamming distance on $\Sigma_H$. Along the unique path between two labels $\sigma_H, \tau_H \in \Sigma_H$, each mismatch at a

---

**Algorithm 1:** Tree Structure for TWWL

---

**Input:** $\mathcal{G}$: a set of graphs with the initial labels $l_0$; $H$: the maximum number of the WL iterations
**Output:** Label tree $\mathcal{T}$ rooted at `root`, and mapping $\phi : \Sigma_H \to V(\mathcal{T})$

**Struct** *NODE*            `// Data Structure for Tree Node`
     depth: INTEGER
     $\Gamma$: VECTOR OF $N$ DOUBLES
     children: HASHMAP<LABEL,NODE>
**end**

**Function** *GETORCREATECHILD(*`node`, `label`*)*
     **if** HAS(`node.children`, `label`) **then**
        **return** `node.children[label]`
     **end**
     `child` $\leftarrow$ NODE(`depth=node.depth+1`, `label=label`, $\Gamma$`=`$\mathbf{0}_N$, `children=[]`)
     INSERT(`node.children`, `key=label`, `value=child`)
     **return** `child`
**end**

**1** `root` $\leftarrow$ NODE(`depth=-1`, `label=""`, $\Gamma$`=`$\mathbf{0}_N$, `children=[]`)
**2** `que` $\leftarrow$ empty FIFO Queue⟨NODE, GRAPH ID, VERTEX⟩
   `// process for h=0`
**3 for** $\mathcal{G}_i$ in $\mathcal{G}$ **do**
**4**    **for** $v$ in $V_i$ **do**
**5**      `child` $\leftarrow$ GETORCREATECHILD(`root`, $l_0(v)$)
**6**      ENQUEUE(`que`, (`child`, $i$, $v$))
**7**    **end**
**8 end**
   `// process for h=1,...,H`
**9 while** !ISEMPTY(`que`) **do**
**10**    (`node`, $i$, $v$) $\leftarrow$ DEQUEUE(`que`)
**11**    `node.`$\Gamma_i$ $\leftarrow$ `node.`$\Gamma_i$ $+ 1/n_i$
**12**    **if** `node.depth < H` **then**
**13**      $h \leftarrow$ `node.depth`
**14**      `child` $\leftarrow$ GETORCREATECHILD(`node`, $l_{h+1}(v)$)
**15**      ENQUEUE(`que`, (`child`, $i$, $v$))
**16**    **end**
**17 end**
**18 return** `root`

---

level contributes two unit-length edges, so assigning length $1/2$ to each edge yields a total path length equal to the number of mismatched levels, i.e., $d_{\mathrm{Ham}}(\sigma_H, \tau_H) = d_{(\mathcal{T},1/2)}(\sigma_H, \tau_H)$. Accordingly, the factor $1/2$ in Equation 5 reflects this result.

In lines 9–17 of Algorithm 1, each dequeue operation can trigger at most one enqueue operation, during which the depth is incremented by one. Hence, the total number of operations in this part is bounded by the product of the initial queue size and the maximum depth $H$. Since the initial queue elements are inserted one per iteration in lines 3–8, their number is $\sum_{i=1}^N n_i = n$. Moreover, the time complexity of search and insertion in a hash map is $O(1)$ on average. Therefore, the overall time complexity of Algorithm 1 is $O(Hn)$.

The above results imply that the Wasserstein distance for the categorical WWL graph kernel can be computed efficiently as follows.

**Proposition 1.** *Suppose that $\mu, \nu$ are probability measures supported on $\Sigma_H$. Let $(\mathcal{T}, d_{\mathcal{T}}, \phi)$ denote a label tree, path-length metric, and mapping from $\Sigma_H$ to $\mathcal{T}$ generated by Algorithm 1, respectively. Then, the*

*Wasserstein distance in Equation 3 can be written as*

$$W_{d_{Ham}}(\mu, \nu) = \sum_{e \in E(\mathcal{T})} \frac{1}{2} |\phi_\sharp \mu(\Gamma(v_e)) - \phi_\sharp \nu(\Gamma(v_e))|, \tag{5}$$

*where we let $\phi_\sharp \mu$ denote the push-forward measure of $\mu$ by $\phi$.*

*Proof.* Since $W_{d_{\mathcal{T}}}(\phi_\sharp \mu, \phi_\sharp \nu) = \sum_{e \in E(\mathcal{T})} \frac{1}{2} |\phi_\sharp \mu(\Gamma(v_e)) - \phi_\sharp \nu(\Gamma(v_e))|$ holds by Theorem 1, we show that $W_{d_{\mathrm{Ham}}}(\mu, \nu) = W_{d_{\mathcal{T}}}(\phi_\sharp \mu, \phi_\sharp \nu)$.

First, let $\pi$ be a coupling of $\mu$ and $\nu$. Note that $\phi$ is injective and measurable by Lemma 4, the mapping $\phi \times \phi$ is a measurable from $\Sigma_H^2$ to $\mathcal{T}^2$, and $(\phi \times \phi)_\sharp \pi$ is a push-forward measure on $\mathcal{T}^2$ that is also a coupling of $\phi_\sharp \mu$ and $\phi_\sharp \nu$. Since the mapping $\phi$ is distance-preserving, we have

$$\int_{\Sigma_H^2} d_{\mathrm{Ham}}(x, y)\pi(dx, dy) = \int_{\mathcal{T}^2} d_{\mathcal{T}}(a, b)((\phi \times \phi)_\sharp \pi)(da),$$

$$\Rightarrow \inf_{\pi \in \Pi(\mu, \nu)} \int_{\Sigma_H^2} d_{\mathrm{Ham}}(x, y)\pi(dx, dy) \geq \inf_{\gamma \in \Pi(\phi_\sharp \mu, \phi_\sharp \nu)} \int_{\mathcal{T}^2} d_{\mathcal{T}}(a, b)\gamma(da, db) = W_{d_{\mathcal{T}}}(\phi_\sharp \mu, \phi_\sharp \nu).$$

Therefore, it follows that $W_{d_{\mathrm{Ham}}}(\mu, \nu) \geq W_{d_{\mathcal{T}}}(\phi_\sharp \mu, \phi_\sharp \nu)$.

Next, we show that $W_{d_{\mathrm{Ham}}}(\mu, \nu) \leq W_{d_{\mathcal{T}}}(\phi_\sharp \mu, \phi_\sharp \nu)$. Since the map $\phi$ restricted to its image $\phi(\Sigma_H)$ is bijective, its inverse map $\phi^{-1} : \phi(\Sigma_H) \to \Sigma_H$ exists, and $\phi^{-1}$ is distance-preserving.

Let $\gamma$ denotes a coupling of $\phi_\sharp \mu$ and $\phi_\sharp \nu$, then

$$\int_{\mathcal{T}^2} d_{\mathcal{T}}(a, b)\gamma(da, db) \geq \int_{\phi(\Sigma_H)^2} d_{\mathcal{T}}(a, b)\gamma(da, db), \quad \text{since } \phi^{-1} \text{ is distance-preserving,}$$

$$= \int_{\Sigma_H^2} d_{\mathrm{Ham}}(x, y)((\phi^{-1} \times \phi^{-1})_\sharp \gamma)(dx, dy).$$

$((\phi^{-1} \times \phi^{-1})_\sharp \gamma)$ is a measure on $\Sigma_H^2$, and is a coupling of $\mu$ and $\nu$. Consequently, it leads to

$$\inf_{\gamma \in \Pi(\phi_\sharp \mu, \phi_\sharp \nu)} \int_{\mathcal{T}^2} d_{\mathcal{T}}(a, b)\gamma(da, db) \geq \inf_{\pi \in \Pi(\mu, \nu)} \int_{\Sigma_H^2} d_{\mathrm{Ham}}(x, y)\pi(dx, dy) = W_{d_{\mathrm{Ham}}}(\mu, \nu).$$

This means $W_{d_{\mathcal{T}}}(\phi_\sharp \mu, \phi_\sharp \nu) \geq W_{d_{\mathrm{Ham}}}(\mu, \nu)$ holds. $\qquad\square$

The positive definiteness proof of the categorical WWL kernel in the original WWL paper (Togninalli et al., 2019) already hints at a hierarchical structure of the Wasserstein computation. The proof shows that, under a Hamming ground cost, the Wasserstein distance between WL embeddings can decompose across the depth of the WL iterations, and the optimal coupling at the final layer induces optimal couplings at earlier layers. In this sense, a hierarchical structure is implicitly present. Proposition 1 makes this structure explicit by identifying the label tree. By treating the WL embeddings as probability measures on the tree, we show that applying Theorem 1 derives a closed-form expression for the Wasserstein distance.

This result shows that the TWWL algorithm exhibits lower time complexity than the approach described in Togninalli et al. (2019). Given two labeled graphs $\mathcal{G}_1$ and $\mathcal{G}_2$ with $n_1$ and $n_2$ vertices, respectively, the time complexity of the Sinkhorn algorithm for Equation 3 is $\mathcal{O}(n_1 n_2)$. The time complexity of the TWWL algorithm is determined by the number of edges in the label tree and the computation of $\Gamma$. The number of edges is $\mathcal{O}(n_1 + n_2)$ even in the worst case, and $\Gamma$ can be computed in $\mathcal{O}(n_1 + n_2)$ time beforehand. Consequently, the overall time complexity of our algorithm is $\mathcal{O}(n_1 + n_2)$. As the number of vertices in the given graphs increases, our algorithm becomes significantly faster than the existing approach.

Equation 4 has been shown to be positive definite for every $\lambda > 0$ by Togninalli et al. (2019). The alternative expression based on the Tree Wasserstein distance offers an independent verification. From Proposition 2 in Le et al. (2019b, Proposition 2), it follows that $W_{d_{\mathrm{Ham}}}(\mu, \nu)$, expressed in Equation 5, is negative definite. Furthermore, Theorem 2.2 in Berg et al. (1984) proves that the graph kernel given in Equation 4 is positive definite.

Table 1: Summary of datasets. We denote by $N$ the number of graphs in the dataset, and by $\bar{n}$ and $\mathrm{avg}\{|E_i|\}$ the average number of vertices and edges per graph, respectively.

| | $N$ | $\bar{n}$ | $\mathrm{avg}\{|E_i|\}$ | ♯graphs per class | node labels |
|---|---|---|---|---|---|
| MUTAG | 188 | 17.93 | 19.79 | 63/125 | ✓ |
| PTC-MR | 344 | 14.29 | 14.69 | 192/152 | ✓ |
| ENZYMES | 600 | 32.63 | 62.14 | 100/100/100/100/100/100 | ✓ |
| PROTEINS | 1113 | 39.06 | 72.82 | 663/450 | ✓ |
| DD | 1178 | 284.32 | 715.66 | 691/487 | ✓ |
| NCI1 | 4110 | 29.87 | 32.30 | 2053/2057 | ✓ |
| COLLAB | 5000 | 74.49 | 2457.78 | 2600/775/1625 | - |
| REDDIT-BINARY | 2000 | 429.62 | 497.75 | 1000/1000 | - |
| REDDIT-MULTI-5K | 4999 | 508.51 | 594.87 | 1000/1000/1000/1000/999 | - |
| REDDIT-MULTI-12K | 11929 | 391.40 | 456.89 | 767/2592/1000/1094/902/1205 513/999/1243/1092/522 | - |
| DBLP-v1 | 19456 | 10.48 | 19.65 | 9530/9926 | ✓ |
| github-stargazers | 12725 | 113.79 | 234.64 | 5917/6808 | - |

## 5 Experiments

In this section, we conduct two numerical experiments to validate the scalability and performance of our approach. The experimental code is publicly available at `https://github.com/KeishiS/twwl`.

We benchmark our algorithm against the Wasserstein Weisfeiler-Lehman kernel with the Sinkhorn algorithm (Togninalli et al., 2019, WWL), the Weisfeiler-Lehman subtree kernel (Shervashidze et al., 2011, WL), the Weisfeiler-Lehman Optimal Assignment kernel (Kriege et al., 2016, WL-OA), traditional kernels such as the Shortest Path kernel (Borgwardt & Kriegel, 2005, SP), and Graphlet sampling kernel (Pržulj, 2007, GL), and recent methods such as the Multiscale Laplacian graph kernel (Kondor & Pan, 2016, MLG), the Weisfeiler-Lehman Lower Bound distance (Chen et al., 2022, WLLB)[2], and the Relaxed Weisfeiler-Lehman kernel (Schulz et al., 2022, R-WL)[3]. For WL, WL-OA, SP, GL and MLG, we utilize the graph kernel implementations provided by Siglidis et al. (2020). All experiments are conducted on an Ubuntu 22.04 machine equipped with an Intel Xeon Gold 6354 CPU and 256GB of RAM. For runtime comparisons against WWL, both our method and the WWL kernel are implemented entirely in Julia to ensure fairness.

We evaluate the proposed methods on graph classification task using real-world datasets summarized in Table 1. For datasets without node labels, we assign degree-based dummy labels. Across baselines, node labels are used as follows. WL, WL-OA and WWL use them as inputs to the WL algorithm. R-WL represents WL labels as unfolding trees and assesses their similarity via a tree edit metric. SP incorporates them when comparing two shortest paths by taking into account the agreement of node labels at the endpoints of corresponding edges along the paths. In WLLB, node labels instantiate the ground label space from which the hierarchy of probability measures is built. MLG uses node label agreement as the base kernel over vertex features. GL ignores node labels and measures similarity via counts of graphlet patterns. All datasets are downloaded from Morris et al. (2020). The details on experiments are described in Appendix B.

### 5.1 Runtime Comparison

In the first experiment, we benchmark the computation time required for calculating the pairwise Wasserstein distances on real-world datasets. For each combination of datasets and methods, we conduct 10 independent trials and report the average runtime (in seconds) and the corresponding standard deviation. We denote linear programming solver by LP, implemented by Huangfu & Hall (2018), and note that our algorithm provides the same calculated values with the solver. Furthermore, since the computation time of the Sinkhorn

---

[2]`https://github.com/chens5/WL-distance`
[3]`https://github.com/mlai-bonn/GenWL`

Table 2: Runtime performance (in seconds) of the Wasserstein distance computation fixed at $H = 5$. "NA (not-available)" indicates timeout (over 24h).

| METHOD | MUTAG | PTC-MR | ENZYMES | PROTEINS | DD | NCI1 |
|---|---|---|---|---|---|---|
| LP | $33.89 \pm 1.68$ | $85.98 \pm 5.73$ | $882.56 \pm 69.52$ | $4597.12 \pm 157.77$ | NA | $34\,775.82 \pm 337.00$ |
| WWL (0.01) | $3.15 \pm 0.04$ | $4.54 \pm 0.01$ | $37.01 \pm 0.02$ | $147.26 \pm 0.52$ | $9390.61 \pm 28.49$ | $2116.45 \pm 6.92$ |
| WWL (1.00) | $0.27 \pm 0.01$ | $0.47 \pm 0.00$ | $5.33 \pm 0.16$ | $24.89 \pm 0.23$ | $1283.43 \pm 9.48$ | $229.97 \pm 0.59$ |
| TWWL | $0.03 \pm 0.00$ | $0.08 \pm 0.00$ | $0.67 \pm 0.12$ | $2.38 \pm 0.19$ | $26.93 \pm 0.40$ | $24.33 \pm 0.56$ |

| | COLLAB | **REDDIT-B** | **REDDIT-M5** | **REDDIT-M12** | DBLP-v1 | **github-stargazers** |
|---|---|---|---|---|---|---|
| LP | NA | NA | NA | NA | NA | NA |
| WWL (0.01) | $7721.67 \pm 37.48$ | NA | NA | NA | $1798.11 \pm 5.92$ | NA |
| WWL (1.00) | $1653.68 \pm 27.69$ | $8457.94 \pm 469.58$ | NA | NA | $945.74 \pm 8.84$ | $25\,820.89 \pm 65.90$ |
| TWWL | $44.18 \pm 0.32$ | $38.92 \pm 0.32$ | $333.53 \pm 3.02$ | $5535.76 \pm 25.07$ | $328.89 \pm 0.56$ | $952.02 \pm 2.11$ |

algorithm varies depending on the entropic regularization term $\gamma$, we assess the time for the Sinkhorn algorithm with both 0.01 and 1.0. The parameter $H$ for the WL algorithm is fixed at 5 for simplicity. For further details, additional runtime results for the proposed method with varying values of $H$ are placed in Appendix B.3. We note that the runtime in Table 2 includes only the time taken to compute the pairwise distance matrix, excluding the preprocessing time for LP, WWL, and TWWL. This is because Algorithm 1 used in TWWL as preprocessing is a slight modification of the WL algorithm used for preprocessing in LP and WWL, and its running time is comparable to, or at most slightly slower than that of the WL algorithm. Further runtime decomposition is described in Appendix B.4.

Table 2 shows that TWWL significantly reduces runtime compared to either the Sinkhorn algorithm or linear programming solver. Notably, TWWL completes computations within practical time limits even in challenging scenarios, such as a dataset with numerous graphs (e.g., DBLP-v1), a dataset with high average vertex counts per graph (e.g., REDDIT-MULTI-5K), and a dataset with both features (e.g., REDDIT-MULTI-12K). For some datasets, such as NCI1, DD, and larger datasets, it was frequently observed that the Sinkhorn algorithm failed to converge within the maximum number of iterations for any value of $\gamma$. A key advantage of our algorithmic approach is its reliability to produce stable solutions.

## 5.2 Performance Comparison

In this experiment, we evaluate the performance of TWWL on a graph classification task using 10-fold cross-validation with a support vector machine classifier (Cortes & Vapnik, 1995). All hyperparameters are selected via grid search on the training folds only: the SVM regularization parameter, the number of WL iterations $H$, the kernel scale $\lambda$, and the Sinkhorn entropic regularization term in WWL. Additional details on the hyperparameter settings are given in Appendix B.2. Since the TWWL algorithm is specifically designed to accurately and efficiently compute the Wasserstein distance, it is expected that our algorithm achieves comparable performance to the WWL kernel. Our aim is to verify that the precise Wasserstein distance does not adversely affect performance when compared to the Sinkhorn algorithm.

Table 3 shows that our algorithm achieves classification accuracy competitive with the WWL kernel. Notably, TWWL can scale to large datasets for which the Sinkhorn algorithm is computationally prohibited due to the runtime and/or memory requirements. This result suggests that our proposed approach can serve as a drop-in replacement for the Sinkhorn algorithm in the WWL kernel in the context of categorical node labels.

## 6 Conclusion

In this paper, we proposed an exact and efficient algorithm to compute the Wasserstein Weisfeiler-Lehman graph kernel for datasets with categorical node labels, especially for large-scale application domains. We revealed that labels generated by the Weisfeiler-Lehman algorithm inherently yield a tree structure, and that the proposed algorithm provides the same result as the original WWL graph kernel. The computational complexity of the proposed algorithm scales linearly with the number of vertices in a graph, which is considerably more efficient than the quadratic computational cost required by the Sinkhorn algorithmic approach. Numerical experiments demonstrate that our algorithm achieves performance comparable to the Sinkhorn

Table 3: Classification accuracies on datasets. "NA" and "OOM" indicate timeout (over 24h) and out-of-memory (over 256GB), respectively.

| METHOD | MUTAG | PTC-MR | ENZYMES | PROTEINS | DD | NCI1 |
|---|---|---|---|---|---|---|
| SP | $81.84 \pm 8.62$ | $61.00 \pm 6.75$ | $40.67 \pm 6.54$ | $\mathbf{75.74 \pm 2.75}$ | $\mathbf{79.80 \pm 2.37}$ | $73.26 \pm 1.95$ |
| GL | $77.02 \pm 9.70$ | $55.82 \pm 4.15$ | $29.83 \pm 5.24$ | $68.92 \pm 3.45$ | $72.49 \pm 4.52$ | $59.61 \pm 2.35$ |
| WL | $85.00 \pm 10.51$ | $62.27 \pm 11.90$ | $54.33 \pm 6.95$ | $75.65 \pm 2.86$ | $79.20 \pm 3.66$ | $85.18 \pm 1.51$ |
| WL-OA | $85.56 \pm 6.92$ | $62.24 \pm 5.88$ | $\mathbf{60.67 \pm 5.62}$ | $75.21 \pm 2.58$ | $79.46 \pm 2.29$ | $86.16 \pm 1.35$ |
| WWL | $87.72 \pm 6.91$ | $65.12 \pm 7.27$ | $60.00 \pm 5.50$ | $74.58 \pm 2.54$ | NA | $86.37 \pm 1.40$ |
| MLG | $\mathbf{88.30 \pm 9.21}$ | $57.82 \pm 7.55$ | $60.17 \pm 6.73$ | $75.65 \pm 2.55$ | NA | $80.88 \pm 2.42$ |
| R-WL | $87.69 \pm 8.07$ | $54.38 \pm 7.88$ | $45.33 \pm 5.65$ | $74.67 \pm 3.89$ | NA | $78.73 \pm 1.34$ |
| WLLB | $88.27 \pm 8.29$ | $53.19 \pm 8.82$ | $37.83 \pm 6.76$ | NA | NA | NA |
| TWWL | $\mathbf{88.27 \pm 6.16}$ | $\mathbf{66.00 \pm 6.83}$ | $59.83 \pm 4.74$ | $74.94 \pm 2.98$ | $77.17 \pm 3.51$ | $\mathbf{86.57 \pm 1.27}$ |

| | COLLAB | REDDIT-B | REDDIT-M5 | REDDIT-M12 | DBLP-v1 | github-stargazers |
|---|---|---|---|---|---|---|
| SP | $\mathbf{81.34 \pm 2.47}$ | $\mathbf{88.70 \pm 1.93}$ | $52.33 \pm 2.62$ | $\mathbf{44.10 \pm 1.14}$ | OOM | $\mathbf{68.06 \pm 1.17}$ |
| GL | NA | NA | NA | NA | NA | NA |
| WL | $78.74 \pm 2.51$ | $75.05 \pm 2.07$ | $50.79 \pm 1.46$ | $39.65 \pm 1.33$ | $92.97 \pm 0.52$ | $65.19 \pm 1.03$ |
| WL-OA | NA | $88.60 \pm 2.33$ | NA | NA | NA | NA |
| WWL | NA | NA | NA | NA | OOM | NA |
| MLG | NA | NA | NA | OOM | OOM | NA |
| R-WL | NA | NA | NA | NA | OOM | NA |
| WLLB | NA | NA | NA | NA | NA | NA |
| TWWL | $80.28 \pm 1.67$ | $85.75 \pm 2.87$ | $\mathbf{54.47 \pm 2.48}$ | $42.90 \pm 0.95$ | $\mathbf{93.37 \pm 0.47}$ | $65.55 \pm 0.73$ |

algorithm, and its efficient computation enables the WWL kernel to be applied to large-scale datasets that were previously infeasible, thereby establishing it as an alternative for datasets with categorical node labels.

As our goal was a faithful and scalable realization of WWL with the Hamming ground cost, we fixed all edge lengths in the label tree to $1/2$. This choice uniquely preserves exact equivalence between the tree path metric and the Hamming metric over WL label sequences. Allowing heterogeneous per-edge weights would define a different ground metric over label sequences and thus a different kernel family. Exploring data-dependent tree metrics is orthogonal to our focus here and constitutes promising future work.

Another promising direction for future work is to extend the proposed approach to datasets with continuous node attributes, e.g., using a clustering approach, as in the bag-of-visual-words approach in computer vision. Our method is based on the property that the space formed by the label set and the Hamming distance can be represented as a tree structure while preserving its topological properties. However, this property does not generally hold for continuous data, necessitating approximations. One possible approach is to develop a graph kernel based on the tree-sliced Wasserstein distance (Le et al., 2019b). While existing work (Perez et al., 2024) accelerates the WWL kernel by leveraging the sliced Wasserstein distance, a systematic comparison of these strategies, focusing on both computational efficiency and performance, would be valuable. In addition, there have been numerous studies on approximating Euclidean structures with trees, an investigation into optimal tree construction strategies may yield informative insights and enhance the effectiveness of our approach for continuous data.

### Acknowledgements

We thank anonymous reviewers for insightful comments and suggestions. Part of this work is supported by JSPS KAKENHI 23K24909 and 25H01494. TL gratefully acknowledges the support of JSPS KAKENHI Grant number 23K11243, and Mitsui Knowledge Industry Co., Ltd. grant.

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

## A  Hash Functions

In many areas of computer science, a hash function is used to map arbitrary length data (e.g., text data, image, or binary data) to a fixed length. This output, commonly called a hash value, serves as a compact representation of the original data. In general, hash functions are used for two main purposes. One is for data uniqueness verification and tamper detection in cryptographic security, and the other is as table keys for processing large volumes of data or for detecting differences between two data in non-cryptographic domains. SHA-2 (Hansen & 3rd, 2011) is one of the well-known standards designed by the National Security Agency and standardized by the National Institute of Standards and Technology as a hash function for cryptographic security.

The properties required of a hash function depend on its intended application, but to be considered a good hash function, it must satisfy at least the following two properties:

- Deterministic: for the same input, the hash function always produces the same hash value.

- Collision resistance: it should be computationally infeasible to find two different inputs that yield the same hash value.

The hash function in Equation 1 is assumed to satisfy these two properties. A well-known SHA-256 implementation can be used to compute Equation 1. Alternatively, Julia's built-in `Base.hash` function is used in our implementation.

## B  Details on Experiments

### B.1  Dataset Descriptions

Every graph dataset contains the adjacency information and a graph-level label. Some datasets additionally provide optional information such as node labels/attributes or edge labels/attributes. In our experiments, we exploited only the node-label information when it was available. For datasets without node labels (COLLAB, REDDIT-BINARY, REDDIT-MULTI-5K, REDDIT-MULTI-12K, github-stargazers), we assigned pseudo node labels based on its degree.

Datasets used in our experiments can be categorized into two types: chemoinformatics and bioinformatics datasets, and social network datasets. The main characteristics of each dataset are summarized as follows.

*MUTAG*: This is a chemoinformatics dataset consisting of 188 aromatic and heteroaromatic compounds (Debnath et al., 1991). Each graph models a compound, with nodes corresponding to atoms and edges to the chemical bonds between them. The task is to predict the mutagenic effect of the compound, categorized into two classes.

*PTC-MR*: This dataset originates from the Predictive Toxicology Challenge (Helma et al., 2001, PTC), a competition to promote the development of machine learning models for predicting chemical toxicity. The PTC-MR dataset contains chemical compounds tested for carcinogenicity in male rats (MR). Each compound is modeled as a graph, where nodes denote atoms and edges denote the chemical bonds connecting them. The task is to predict compound carcinogenicity, which is a binary classification problem.

*ENZYMES*: This is a bioinformatics dataset constructed from the BRENDA (Braunschweig Enzymes Database), comprehensive enzymes and metabolic information repository (Borgwardt et al., 2005). Each graph represents an enzyme, where nodes are its secondary structure elements (SSEs), and edges are created between nodes if they are neighbors along the amino acid sequence or are spatially close. The task is to classify enzymes into six classes based on their catalytic activity.

*PROTEINS*: This dataset is used as a benchmark for graph classification task of predicting whether a protein is an enzyme or not (Borgwardt et al., 2005). A protein is represented by a graph as follows. Nodes in the graph correspond to SSEs, and they are connected by an edge if they are neighbors along the amino acid sequence or are spatially close.

*DD*: This dataset consists of 1178 protein structures where the task is to classify them into enzymes and non-enzymes (Dobson & Doig, 2003). Each graph represents a protein, in which the nodes are amino acids and two nodes are connected by an edge if they are less than 6 Angstroms apart.

*NCI1*: This dataset consists of chemical compounds screened for their effectiveness against non-small cell lung cancer, with the task being to classify each compound as active or not (Wale et al., 2008). Nodes in each graph represent atoms, and edges represent chemical bonds between them.

*COLLAB*: This is a scientific collaboration dataset of researchers (Leskovec et al., 2007). Each graph represents an ego-network of a researcher. In the graph, nodes represent researchers and edges indicate co-authorship. This dataset is widely used as a benchmark for graph classification tasks of predicting the researcher's academic field among three classes: High Energy Physics, Condensed Matter Physics, and Astro Physics.

*REDDIT-BINARY/MULTI-5K/MULTI-12K*: These are social network dataset created from Reddit, an online discussion platform where users participate in discussions within topic-specific threads called subreddits (Yanardag & Vishwanathan, 2015). Each graph represents the user interaction structure of a single discussion thread. In each graph, nodes represent users, and an edge connects two nodes if one user replied to another's comment. The collection includes three variants with different classification challenges:

- *REDDIT-BINARY* is a binary classification task to distinguish threads from QA-style subreddits versus discussion-style subreddits.

- *REDDIT-MULTI-5K* is a 5-class classification task among five subreddits: worldnews, videos, AdviceAnimals, aww and mildlyinteresting.

- *REDDIT-MULTI-12K* expands the task to a larger set of subreddits: AskReddit, AdviceAnimals, atheism, aww, IAmA, mildlyinteresting, Showerthoughts, videos, todayilearned, worldnews, and TrollXChromosomes.

*DBLP-v1*: This is a scientific collaboration dataset derived from DBLP, a comprehensive bibliography database of computer science publications (Pan et al., 2013). Each graph represents a paper in DBLP, where nodes denote either a paper or a keyword, and edges represent one of three types of relationships: a citation relationship between two papers, a link between a paper and its corresponding keyword, or a link between keywords from the same paper. The task is to classify each paper into one of two classes: database and data mining field or computer vision and pattern recognition field.

*github-stargazers*: This is a social network dataset derived from GitHub, the software development platform (Rozemberczki et al., 2020). It consists of social networks of developers who starred popular machine learning and web development repositories. Each graph represents the community of developers who starred a repository, where nodes represent users, and edges denote follower relationships. The goal is to predict whether a social network belongs to a machine learning or web development repository.

## B.2 Hyperparameter Settings

The following hyperparameter settings were used in the experiments of Section 5. For a regularization parameter $C$ in the support vector classifier, we search over the range $\{10^{-3}, 10^{-2}, \ldots, 10^3\}$. The number of iterations $H$ for the WL algorithm is chosen from $\{1, \ldots, 7\}$. The parameter $\lambda$ of the WWL and TWWL is selected from $\{10^{-4}, 10^{-3}, \ldots, 10^1\}$. For the Sinkhorn algorithm, we select the entropic regularization parameter from $\{0.01, 0.05, 0.1, 0.2, 0.5, 1, 10\}$ and fix the maximum number of iterations to 1000, which is the default setting in the commonly used implementation (Flamary et al., 2021). In the second experiment, hyperparameter optimization is performed on the training data via grid search over the predetermined range of values.

Table 4: TWWL runtime performance of the Wasserstein distance computation as the parameter $H$ is increased.

| DATASET | $H=1$ | $H=2$ | $H=3$ | $H=4$ | $H=5$ | $H=6$ | $H=7$ |
|---|---|---|---|---|---|---|---|
| MUTAG | $0.007 \pm 0.004$ | $0.010 \pm 0.001$ | $0.015 \pm 0.001$ | $0.022 \pm 0.001$ | $0.032 \pm 0.001$ | $0.042 \pm 0.001$ | $0.052 \pm 0.001$ |
| PTC-MR | $0.036 \pm 0.066$ | $0.040 \pm 0.057$ | $0.038 \pm 0.001$ | $0.056 \pm 0.001$ | $0.076 \pm 0.001$ | $0.096 \pm 0.001$ | $0.181 \pm 0.213$ |
| ENZYMES | $0.08 \pm 0.07$ | $0.20 \pm 0.06$ | $0.36 \pm 0.09$ | $0.51 \pm 0.09$ | $0.70 \pm 0.16$ | $0.84 \pm 0.11$ | $1.04 \pm 0.15$ |
| PROTEINS | $0.24 \pm 0.09$ | $0.64 \pm 0.10$ | $1.19 \pm 0.16$ | $1.83 \pm 0.14$ | $2.37 \pm 0.16$ | $2.98 \pm 0.23$ | $3.47 \pm 0.18$ |
| DD | $6.20 \pm 0.31$ | $11.61 \pm 0.47$ | $16.67 \pm 0.43$ | $21.74 \pm 0.17$ | $26.72 \pm 0.18$ | $31.83 \pm 0.28$ | $36.96 \pm 0.36$ |
| NCI1 | $5.47 \pm 0.45$ | $10.19 \pm 0.53$ | $15.56 \pm 0.54$ | $20.70 \pm 0.38$ | $25.22 \pm 0.49$ | $30.20 \pm 0.30$ | $34.81 \pm 0.31$ |
| COLLAB | $18.08 \pm 0.53$ | $25.97 \pm 0.85$ | $34.77 \pm 0.63$ | $43.69 \pm 0.56$ | $52.24 \pm 0.66$ | $60.90 \pm 0.71$ | $69.66 \pm 0.58$ |
| REDDIT-B | $5.50 \pm 0.14$ | $13.55 \pm 0.14$ | $22.36 \pm 0.32$ | $30.62 \pm 0.57$ | $38.74 \pm 0.45$ | $47.08 \pm 0.54$ | $55.27 \pm 0.49$ |
| REDDIT-M5 | $66.58 \pm 1.29$ | $138.67 \pm 3.30$ | $207.05 \pm 2.08$ | $388.21 \pm 53.25$ | $554.52 \pm 2.83$ | $686.86 \pm 2.78$ | $817.82 \pm 1.12$ |
| REDDIT-M12 | $326.64 \pm 1.28$ | $835.47 \pm 1.66$ | $1476.17 \pm 3.76$ | $2155.87 \pm 6.07$ | $5631.26 \pm 16.50$ | $7246.46 \pm 32.74$ | $8688.33 \pm 30.20$ |
| DBLP-v1 | $139.52 \pm 0.85$ | $187.52 \pm 0.69$ | $236.03 \pm 0.93$ | $285.69 \pm 1.27$ | $333.33 \pm 0.75$ | $382.69 \pm 1.34$ | $431.27 \pm 0.85$ |
| github-stargazers | $208.34 \pm 0.19$ | $398.80 \pm 0.54$ | $598.15 \pm 0.52$ | $794.32 \pm 1.12$ | $987.71 \pm 1.21$ | $1184.01 \pm 1.37$ | $1377.23 \pm 3.00$ |

### B.3 Further results on runtime comparison

To evaluate how the computational cost of our algorithm scales with the parameter $H$, we supplement the runtime study in Section 5.1 with additional experiments. Whereas the experiment in Section 5.1 fixed $H = 5$, Table 4 reports the runtime results of the TWWL algorithm for a range of $H$ values. In the TWWL algorithm, the extra cost caused by increasing $H$ is proportional to the number of newly generated labels, whose maximum is bounded by the total number of vertices $n$ in a dataset. Consequently, the overall complexity grows approximately linearly in $H$, a trend clearly confirmed by Table 4.

### B.4 Additional results on Runtime comparison

This appendix augments the runtime comparison in Section 5.1 by reporting not only preprocessing for WWL and TWWL but also, for reference, the end-to-end runtimes of existing methods. For categorical node labels, WWL uses the WL algorithm for preprocessing and then computes the Wasserstein distance in Equation 3. In contrast, the TWWL algorithm first constructs a label tree according to Algorithm 1 as preprocessing, and subsequently evaluates the Tree Wasserstein distance in Equation 5 based on that tree. Table 2 compares the running times of Equation 3 for WWL and Equation 5 for TWWL, and does not include preprocessing times. In this appendix, we summarize the end-to-end runtimes for computing the Wasserstein distance for WWL and TWWL, and we also report, for reference, the time required to construct the Gram matrix for existing methods. We note that WWL and TWWL are implemented in Julia. For the other methods, we use the authors' publicly available Python implementations on GitHub, so absolute runtimes are not strictly comparable and should be interpreted only as indicative.

In Table 5, we report the end-to-end time required to construct the Gram matrix. Algorithm 1 makes only minor modifications to the WL algorithm and adds the steps required to construct the tree structure. As shown in Table 5, the preprocessing time of TWWL is at most twice that of WWL across datasets.

We note that the entropic regularization parameter in WWL does not affect WL preprocessing, and the preprocessing required for the linear programming approach is identical to that of WWL. We therefore omitted these cases from Table 5.

Table 5: Runtime decomposition of the WWL and TWWL algorithms fixed at $H = 5$, and end-to-end runtimes of comparison methods in seconds. "NA" and "OOM" indicate timeout (over 24h) and out-of-memory (over 256GB), respectively. Entries for WWL and TWWL are reported as a pair: the mean preprocessing time (left) and the mean main computation (right).

| METHOD | MUTAG | PTC-MR | ENZYMES | PROTEINS | DD | NCI1 |
|---|---|---|---|---|---|---|
| WWL (1.00) | 0.10 / 0.27 | 0.02 / 0.47 | 0.12 / 5.33 | 0.24 / 24.89 | 2.46 / 1283.43 | 0.75 / 229.97 |
| TWWL | 0.02 / 0.03 | 0.03 / 0.08 | 0.16 / 0.67 | 0.40 / 2.38 | 4.42 / 26.93 | 1.15 / 24.33 |
| SP | 0.15 | 0.24 | 1.69 | 8.32 | 715.17 | 11.71 |
| WL | 0.08 | 0.11 | 0.52 | 0.93 | 7.23 | 3.37 |
| WL-OA | 0.04 | 0.06 | 0.34 | 0.73 | 10.61 | 4.20 |
| GL | 0.34 | 0.45 | 7.35 | 12.76 | 236.91 | 12.39 |
| MLG | 28.64 | 45.73 | 197.99 | 758.41 | NA | 1486.69 |
| R-WL | 6.69 | 14.59 | 893.46 | 3062.59 | NA | 161.08 |
| WLLB | 867.23 | 1599.38 | 19169.05 | NA | NA | NA |

| | COLLAB | REDDIT-B | REDDIT-M5 | REDDIT-M12 | DBLP-v1 | github-stargazers |
|---|---|---|---|---|---|---|
| WWL (1.00) | 7.38 / 1653.68 | 6.30 / 8457.94 | 20.55 / NA | 45.90 / NA | 1.44 / 945.74 | 12.11 / 25 820.90 |
| TWWL | 12.32 / 44.18 | 10.21 / 38.92 | 32.57 / 333.53 | 70.64 / 5535.76 | 2.71 / 328.89 | 20.59 / 952.02 |
| SP | 1019.29 | 4175.16 | 8598.40 | 14997.43 | OOM | 2464.72 |
| WL | 54.00 | 12.89 | 38.06 | 88.17 | 18.89 | 42.66 |
| WL-OA | NA | 11.83 | NA | NA | NA | NA |
| GL | NA | NA | NA | NA | NA | NA |
| MLG | NA | NA | NA | NA | NA | NA |
| R-WL | NA | NA | NA | NA | NA | NA |
| WLLB | NA | NA | NA | NA | NA | NA |

