# OpenReview forum: "Tree Structure for the Categorical Wasserstein Weisfeiler-Lehman Graph Kernel"
_TMLR — Accepted by TMLR_

### Review · Reviewer_g9jw · 2025-08-14

**Summary Of Contributions:**

In this submission, the authors proposed a tree-structured Wasserstein Weisfeiler-Lehman (TWWL) graph kernel for transductive graph classification. In particular, instead of computing the Wasserstein distance between graphs based on their node-level Hamming distance, the proposed method first extracts a WL-label tree from graphs and then computes the Wasserstein distance equivalently based on the push-forward results derived from the label tree. As a result, the computational complexity is reduced from O(N^2) to O(N) for the graphs with N nodes.

In theory, the authors demonstrate the equivalence between the Wasserstein distance based on the underlying Hamming distance and the proposed TWWL distance.

Experiments show that the proposed TWWL graph kernel method achieves competitive graph classification accuracy compared to existing graph kernel methods. Moreover, it reduces the runtime significantly, showing promising computational efficiency.

**Audience:**

Yes

**Audience Explanation:**

1. Graph learning is an important topic in the machine learning community. Among existing graph learning methods, the graph kernel method corresponds to an interesting and flexible transductive learning paradigm, which is different from the inductive learning paradigm corresponding to learning parametric graph neural networks. Currently, whether the graph kernel method can overcome its natural limitation in computational efficiency is the key problem in this direction.

2. Tree-structured Wasserstein distances have been widely used in machine learning tasks. The proposed method provides a new variant of TWD and demonstrates its equivalence to a special Wasserstein distance, which may attract researchers studying optimal transport-driven machine learning techniques.

**Claims And Evidence:**

Yes

**Claims Explanation:**

1. Tree-structured Wasserstein distance (TWD) has been proposed for a long time and has been widely used as an efficient alternative to the classic Wasserstein distance (WD). The authors proposed a new variant of TWD in the scenario of WL label propagation and demonstrated the equivalence between the proposed TWWL distance and a special WD defined on the Hamming distance of graph nodes.

2. Experimental results show that the proposed method achieves comparable performance with significant improvements in efficiency, verifying a) the theoretical equivalence between the proposed TWWL graph kernel and the WWL one, and b) the superiority of the TWWL graph kernel in computational efficiency.

**Requested Changes:**

1. The differences between the proposed TWWL distance and the existing TWD in Theorem 1 are not explained clearly. I suggest the authors add a new subsection comparing the TWWL distance and the existing TWD methods systematically.

2. In addition, some notations are used without clear definitions, e.g., the $\Gamma$, $w_e$, and $E(\mathcal{T})$ in Theorem 1. In fact, the introduction of TWD is important for understanding the proposed work, especially for readers without sufficient background. Please rewrite this part in detail.

3. In Table 2, I am not sure whether the runtime includes the time of constructing the label tree or not. If the answer is yes, the authors can provide a more detailed runtime analysis, e.g., showing the runtime of different steps of the proposed method and visualizing the runtime as the graph size increases.

4. At the end of section 4, the sentence "The alternative expression based on the Tree Wasserstein distance offers an independent verification through Proposition 2 in Le et al. (2019b)" is too brief. More detailed explanations are necessary.

5. In the experiments, some graphs have node labels. How do the proposed method and baselines leverage the node labels?

6. In the experiments, the baselines are not strong enough. Even if merely considering the graph kernel-based methods, SP and GL are not state-of-the-art, to my knowledge. Some representative and strong baselines, such as the multi-scale Laplacian graph kernel [a] and the fused Gromov-Wasserstein graph kernel [b] should be considered as baselines.

[a] Kondor, Risi, and Horace Pan. "The multiscale laplacian graph kernel." Advances in neural information processing systems 29 (2016).

[b] Titouan, Vayer, et al. "Optimal transport for structured data with application on graphs." International Conference on Machine Learning. PMLR, 2019.

7. In Table 1, "#Classes" should be "#Graphs per class".

---

> ### Author Response · Authors · 2025-09-27
>
> Thank you for your insightful comments. We will make revisions with respect to your feedback as follows.
>
> > ... Among existing graph learning methods, the graph kernel method corresponds to an interesting and flexible transductive learning paradigm, which is different from the inductive learning paradigm corresponding to learning parametric graph neural networks. ...
>
> Thank you for your constructive comments. We apologize for any confusion caused by the accompanying implementation, which may have inadvertently suggested a transductive setting. To clarify, our proposed method is inductive: it supports inference on previously unseen graphs by computing the TWWL kernel between the test graphs and the training data and then applying an SVM for prediction.
>
> > The differences between the proposed TWWL distance and the existing TWD in Theorem 1 are not explained clearly...
>
> We will clarify the connection between the proposed TWWL distance in Proposition 1 and the existing TWD result in Theorem 1. Specifically, after the proof of Proposition 1, we add a remark stating that the TWWL distance is obtained by interpreting the WL embeddings as probability measure on the label tree and applying Theorem 1.
>
> > some notations are used without clear definitions, e.g., the Γ, w_e, and E(Τ) in Theorem 1. In fact, the introduction of TWD is important for understanding the proposed work, especially for readers without sufficient background. Please rewrite this part in detail.
>
> We'll revise the notation section, adding explicit definitions for Γ itself, the node set N(Γ), the edge set E(Γ), and the edge weights w_e. In addition, we expand Section 3.5 to provide a more detailed explanation of the Tree Wasserstein distance.
>
> > In Table 2, I am not sure whether the runtime includes the time of constructing the label tree or not. If the answer is yes, the authors can provide a more detailed runtime analysis, e.g., showing the runtime of different steps of the proposed method and visualizing the runtime as the graph size increases.
>
> The runtime reported in Table 2 does not include the time to construct the label tree. We made this choice because the label tree can be built by augmenting the WL algorithm with a few additional operations. As a result, the preprocessing time is nearly the same between TWWL and WWL, or TWWL slightly slower. For transparency, we will report the measured preprocessing times in Appendix Table 5.
>
> > At the end of section 4, the sentence "The alternative expression based on the Tree Wasserstein distance offers an independent verification through Proposition 2 in Le et al. (2019b)" is too brief. More detailed explanations are necessary.
>
> We revise the end of Section 4 by adding a paragraph that makes explicit the two key theorems: the negative definiteness of the TWD and the result that exp(-λ d) derives a positive definite kernel when d is negative definite.
>
> > In Table 1, "#Classes" should be "#Graphs per class".
>
> We correct Table 1 accordingly.

---

> ### Author Response · Authors · 2025-09-29
>
> > In the experiments, some graphs have node labels. How do the proposed method and baselines leverage the node labels?
>
> We revised the manuscript to clarify the role of node labels. After Lemma 2, we state that TWWL uses node labels to construct the label tree. For the other methods, we describe how node labels are utilized by each method before Section 5.1.
>
> > In the experiments, the baselines are not strong enough. Even if merely considering the graph kernel-based methods, SP and GL are not state-of-the-art, to my knowledge. Some representative and strong baselines, such as the multi-scale Laplacian graph kernel [a] and the fused Gromov-Wasserstein graph kernel [b] should be considered as baselines.
>
> Thank you for your suggestion. We have strengthened the comparisons by adding the Multiscale Laplacian Graph Kernel (MLG), the Relaxed Weisfeiler–Lehman (WL) kernel (as introduced in [1]), and the WL Lower Bound (from [2]). Unfortunately, we were unable to bring the authors’ implementation of the fused Gromov–Wasserstein graph kernel to a runnable state in our environment within the revision timeline. We therefore do not report FGW results at this time, and we apologize for this omission.
>
> [1] Till Hendrik Schulz, Tamás Horváth, Pascal Welke, and Stefan Wrobel. A generalized weisfeiler-lehman graph kernel. Machine learning, 111(7):2601–2629, July 2022. doi: 10.1007/s10994-022-06131-w.
>
> [2] Samantha Chen, Sunhyuk Lim, Facundo Memoli, Zhengchao Wan, and Yusu Wang. Weisfeiler-lehman meets gromov-wasserstein. In Proceedings of the 39th International Conference on Machine Learning, volume 162, pp. 3371–3416, July 2022.

---

### Review · Reviewer_yDmk · 2025-08-14

**Summary Of Contributions:**

This paper proposes the Tree Wasserstein Weisfeiler-Lehman (TWWL) graph kernel, an exact and efficient method for computing the Wasserstein Weisfeiler-Lehman (WWL) kernel for graphs with categorical node labels. The key idea is that the iterative label refinement in the Weisfeiler-Lehman (WL) algorithm naturally produces a hierarchical tree structure over labels, where the path length on this label tree exactly preserves the Hamming distance between label sequences. By leveraging the closed-form formula of the Tree Wasserstein Distance (TWD) on this tree, TWWL reduces the computational complexity of WWL from at least quadratic in the number of nodes to linear, while producing the same kernel values without approximation. Experiments on various real-world graph classification datasets show that TWWL matches WWL’s accuracy but is orders of magnitude faster, enabling the application of Wasserstein-based kernels to large-scale categorical graph datasets that were previously computationally infeasible.

**Audience:**

Yes

**Audience Explanation:**

The paper proposes an efficient method for learning Wasserstein Weisfeiler-Lehman with linear time with regard to the number of nodes. It reduces the time complexity from quadratic time to linear time. I think some people will be interested in it.

**Broader Impact Concerns:**

No ethical concerns.

**Claims And Evidence:**

Yes

**Claims Explanation:**

The submission backs its core claims with (i) clear theory—lemmas showing the WL labels form a tree and that the map from WL labels to the label tree is distance-preserving, which underpins the exact reduction to Tree Wasserstein distance and its closed form (Theorem 1)—and (ii) experiments showing large runtime wins with comparable accuracy to WWL. Theoretical pieces are explicit (Lemma 3, Lemma 4, Proposition 1) and connect the equivalence of the original Wasserstein on Hamming labels to TWD, then argue linear time in total nodes across the two graphs, O(n₁+n₂). Empirically, they describe setup (10 trials for runtimes; 10-fold CV with SVM for accuracy; common baselines) and report consistent order-of-magnitude speedups (e.g., NCI1: 24.33 s vs 229.97–2116.45 s) while matching WWL-style accuracy where WWL runs, and scaling where WWL times out or hits memory limits. Caveats: runtime fairness is partly influenced by fixing H=5 for that study, and while they note additional H sweeps in the appendix, more ablations/significance tests and comparisons to other accelerations (e.g., sliced WWL) would strengthen the evidence; still, for the paper’s stated scope (categorical labels), the evidence is accurate, convincing, and clearly presented.

**Requested Changes:**

1. The concept of tree structures is widely used in many graph learning domains, like [1,2]. What are the differences and correlations between the proposed method and the existing works?

   [1] GFT: Graph Foundation Model with Transferable Tree Vocabulary, NeurIPS 24

   [2] Towards Graph Foundation Models: Learning Generalities Across Graphs via Task-trees, ICML 25

2. It's good to report the time consumption of the proposed method. But I recommend the authors to compare the empirical time consumption between the proposed methods and the existing works.

---

> ### Author Response · Authors · 2025-09-29
>
> Thank you for your careful review and feedback. We will make the following changes to the manuscript based on your comments.
>
> > The concept of tree structures is widely used in many graph learning domains, like [1,2]. What are the differences and correlations between the proposed method and the existing works?
>
> In the context of tree structure, both our method and works such as [1,2] share the goal of efficiently aggregating and representing local information via trees. A key difference is that graph kernels (including ours) explicitly specify the similarity between local neighborhoods through a hand‑designed metric or kernel, whereas GNN/GFM approaches typically learn such representations. Clarifying how the expressivity of kernel‑based tree aggregation relates to that of GNN/GFM is an important direction. The present manuscript does not fully address this perspective, and we leave a thorough investigation to future work.
>
> > It's good to report the time consumption of the proposed method. But I recommend the authors to compare the empirical time consumption between the proposed methods and the existing works.
>
> Thank you for the recommendation. Our method is implemented in Julia whereas many existing baselines are implemented in Python, so timing comparison would be unfair. As we agree that such information is informative, We therefore added reference runtimes for comparison methods in the appendix, together with a note on implementation differences.

---

### Review · Reviewer_M96X · 2025-09-14

**Summary Of Contributions:**

**Summary**: Authors revisit the (categorical) Wasserstein Weisfeiler-Lehman (WWL) kernel between graphs with discrete node features by exposing a tree structure associated to the label set created by the WL procedure called the Label Tree. This structure allows authors to show the equivalence between WWL and the tree Wasserstein distance which admits closed form solutions, called TWWL, that can be computed with linear complexity in the numbers of nodes O(n1 + n2) instead of a cubic complexity for the generic Wasserstein distance. Then they show across a large range of graph datasets that their approach runs in about 3 orders of magnitude faster than a LP solver for the exact Wasserstein and 1 to 2 orders of magnitude for entropically regularized variants. Moreover they show that it achieves  competitive performances with 5 other graph kernels.

**Strengths**:
   - Overall the paper is well written and illustrated.
   - Proofs were carefully checked and I could not find any mistake.
   - Gains in terms of computation times compared to the WWL are significant.

**Weaknesses**:
   - **W1** Authors’ contributions focus on improving the speed of the categorical WWL kernel which has already been outperformed by many other kernels in the recent years such as [A, B, C, D, E, F]. Moreover it is not clear whether TWWL really provides a better trade-off in terms of runtime/performance than these competitors.
   - **W2** The novelty of the method is arguable : The proof of Theorem 1 from the original WWL paper on its positive definiteness already exhibits a tree structure via the OT solution for this problem. Essentially, they show that an OT solution between the WL embeddings is given by an OT solution between embeddings computed at the last iteration. When endowed with a binary cost as done within the Hamming distance, this Wasserstein distance admits a closed form solution given by the L1 distance between masses supported by the finale label set $\Sigma_H$, see Remark 2.25 in Peyré’s book. Therefore it seems that the large part of the theoretical contributions made by authors is a re-framing of a corollary of Theorem 1 of the WWL paper, even if it is nicely done. However, I admit that authors of WWL did not explicit this result, therefore it remains interesting to state it until the end as the current paper is doing.
   - **W3** The method does not support continuous features which are often encountered in real-life applications.
   - **W4** The presentation of Algorithm 1 could be improved with an explanation of the key steps for better readability. Moreover the analysis of the algorithm complexity is not so clear.
   - **W5** Certain small points could be completed or clarified:
       - a) (Intro, page 2) Quantization is also an alternative to approximate the Wasserstein distance with faster computation see e.g [G], note that [E] investigates such alternatives as well as other relaxations to the WWL kernel. Moreover there are also proximal point algorithms to estimate the exact Wasserstein distance which has similar theoretical complexity than Sinkhorn [H]. Finally, it could be beneficial to complete the analysis of the theoretical complexity associated to already mentioned methods to better set the context of the current contribution.
      - b) Equation 2: why the scaling by H as defined in the WWL paper is missing ?
     - c) Could you clarify why a weight ½ is set and not other variants were explored ?
     - d) Proposition 1: It seems to me that you can directly state that $(\Sigma_H, d_{ham})$ is a metric space, why do you formalize that as an assumption ?
     - e)Table 1: Could you explicit notations used in the headers of the table e.g $N$, $\overline{n}$ etc which are not defined nor really consistent.
     - f) Section 5.1: what is the device used for the experiments ?
     - g) Table 3: performances for WWL 0.01 and 1 should be reported.
     - h) The work for Perez & al, 2024 is mentioned several times as a direct competitor and not benchmarked. It should be added to both Table 2 and 3.
    - i) Could you clarify why common solvers for the Wasserstein distance such as the POT library one was not used in this experiment ? I wonder if there are algorithmic details for network flow solvers which make them better than the one used in the benchmark.

[A] Titouan, V., Courty, N., Tavenard, R., & Flamary, R. (2019, May). Optimal transport for structured data with application on graphs. In International Conference on Machine Learning (pp. 6275-6284). PMLR.

[B] Xu, B. C., Ting, K. M., & Jiang, Y. (2021, May). Isolation graph kernel. In Proceedings of the AAAI Conference on Artificial Intelligence (Vol. 35, No. 12, pp. 10487-10495).

[C] Chuang, C. Y., & Jegelka, S. (2022). Tree mover's distance: Bridging graph metrics and stability of graph neural networks. Advances in Neural Information Processing Systems, 35, 2944-2957.


[D] Weisfeiler-lehman meets gromov-Wasserstein. In International Conference on Machine Learning (pp. 3371-3416). PMLR.

[E] Schulz, T. H., Horváth, T., Welke, P., & Wrobel, S. (2022). A generalized weisfeiler-lehman graph kernel. Machine Learning, 111(7), 2601-2629.

[F] Brugère, T., Wan, Z., & Wang, Y. (2024, March). Distances for Markov chains, and their differentiation. In International conference on algorithmic learning theory (pp. 282-336). PMLR.

[G] Chowdhury, S., Miller, D., & Needham, T. (2021, September). Quantized gromov-wasserstein. In Joint European Conference on Machine Learning and Knowledge Discovery in Databases (pp. 811-827). Cham: Springer International Publishing.

[H] Xie, Y., Wang, X., Wang, R., & Zha, H. (2020, August). A fast proximal point method for computing exact wasserstein distance. In Uncertainty in artificial intelligence (pp. 433-453). PMLR.

**Audience:**

Yes

**Audience Explanation:**

The contributions seem relevant for the graph kernel community as the paper introduces an algorithm which seems competitive and able to process very large-scale graphs. However, it could be beneficial the expand the benchmark by including other methods which could operate on such graphs to see if the paper really presents an empirical interest.

**Claims And Evidence:**

Yes

**Claims Explanation:**

Yes on the theoretical side, I checked all the proofs and they seem correct. Authors’ analysis is comprehensive on that matter. Then they also show that their approach indeed improves the computational efficiency of the WWL kernel as claimed in the contributions.

**Requested Changes:**

I suggest to authors to address the weaknesses mentioned above. In particular W4 and W5 should be rather simple to address in order to improve the clarity and comprehensiveness of the paper.
Contributions mentioned in W1 should also be acknowledged in some ways to provide a better review of the literature in the paper and potentially clearly state that some graph kernels were already introduced that achieve better performance than WWL. Clearly many of them have higher computational complexity than TWWL but not all of them so these should be clearly discussed.
Finally, W2 seems to me as a major concern on the necessity of the theoretical contributions to support their methodological contributions.

---

> ### Author Response · Authors · 2025-09-27
>
> We appreciate your constructive feedback. We summarize our responses and will revise as follows.
>
> > W2: The novelty of the method is arguable : The proof of Theorem 1 from the original WWL paper ...
>
> We agree that the original WWL proof implies the presence of a hierarchical structure in the optimal transport solution. Our contribution is to make this latent structure explicit by formalizing the label tree, linking it to the Tree Wasserstein distance, and thereby obtaining a closed-form characterization. To make the boundary between what is implied by the original proof and our novel contribution fully transparent, we will add a paragraph immediately after the proof of Proposition 1 that clarifies this distinction.
>
> > W3: The method does not support continuous features which are often encountered in real-life applications.
>
> We agree that the proposed algorithm does not handle continuous features. To prevent confusion about the scope of applicability, we will revise the manuscript to make this limitation explicit throughout the text.
>
> > W4: The presentation of Algorithm 1 could be improved with an explanation of the key steps for better readability. Moreover the analysis of the algorithm complexity is not so clear.
>
> We'll revise the pseudocode of Algorithm 1 to improve readability by clarifying the key steps and standardizing line numbering in the main part. We also add a complexity analysis for Algorithm 1 immediately after Lemma 4.
>
> > W5-b) Equation 2: why the scaling by H as defined in the WWL paper is missing?
>
> We will add a note immediately after Eq. (2) clarifying that the scaling by the constant H is omitted because it does not affect the results.
>
> > W5-c) Could you clarify why a weight ½ is set and not other variants were explored?
>
> We'll clarify after Lemma 4 that assigning an edge weight of 1/2 on the label tree ensures that the induced path metric exactly matches the Hamming distance between labels. Exploring alternative weights would amount to adopting a different ground metric for labels rather than Hamming. As this lies beyond the scope of the present work, we add this point to the Conclusion as a direction for future research.
>
> > W5-d) Proposition 1: It seems to me that you can directly state that ($Σ_H$, $d_{ham}$) is a metric space, why do you formalize that as an assumption?
>
> We'll correct the description of Proposition 1 as you mentioned.
>
> > W5-e) Table 1: Could you explicit notations used in the headers of the table e.g N, n, etc which are not defined nor really consistent.
>
> We'll update the caption of Table 1 to explicitly state the meaning of each symbol used in the headers.
>
> > W5-f) Section 5.1: what is the device used for the experiments?
>
> We will update Section 5 accordingly to report the device used for the experiments.
>
> > W5-g) Table 3: performances for WWL 0.01 and 1 should be reported.
>
> In our performance comparison, we tune WWL by grid search on the training data, including the entropic regularization term. We add this clarification to Section 5.1.
>
> > W5-i) Could you clarify why common solvers for the Wasserstein distance such as the POT library one was not used in this experiment? I wonder if there are algorithmic details for network flow solvers which make them better than the one used in the benchmark.
>
> We did not use the POT solver because our method is not well suited to dense‑matrix formulations, in short, NumPy‑based implementation would be memory‑inefficient and suboptimal. Since POT relies on highly optimized NumPy/C++ kernels, comparing WWL implementation with POT against a pure‑Python prototype of our method would be unfair. To ensure fairness, we implemented TWWL and WWL in a pure Julia codebase.

---

> ### Author Response · Authors · 2025-09-29
>
> > W1: Authors’ contributions focus on improving the speed of the categorical WWL kernel which has already been outperformed by many other kernels in the recent years such as [A, B, C, D, E, F]. Moreover it is not clear whether TWWL really provides a better trade-off in terms of runtime/performance than these competitors.
>
> Thank you for highlighting these important prior works. As you correctly noted, our goal is to broaden the applicability of the categorical WWL kernel to large-scale graph datasets by accelerating its computation. We have reorganized the Introduction to summarize the characteristics of the relevant methods you mentioned and added text to clarify the positioning of our work in the introduction.
>
> > W5-a: (Intro, page 2) Quantization is also an alternative to approximate the Wasserstein distance with faster computation ... Finally, it could be beneficial to complete the analysis of the theoretical complexity associated to already mentioned methods to better set the context of the current contribution.
>
> Thank you for highlighting these relevant directions. We agree that a more complete discussion of these alternatives and their theoretical complexity would strengthen the paper. In this revision, we have expanded the Introduction’s discussion of fast computation of the Wasserstein distance to summarize and cite these prior approaches. Unfortunately, within the revision timeline, we were unable to conduct a thorough comparative analysis; we leave a systematic theoretical and empirical comparison to future work.
>
> > W5-h) The work for Perez & al, 2024 is mentioned several times as a direct competitor and not benchmarked. It should be added to both Table 2 and 3.
>
> The method of Perez et al. (2024) targets graphs with continuous node attributes, so we do not consider it a direct competitor for our setting, which focuses on graphs with categorical node labels. Instead, we have added comparisons among those you suggested that are designed for, or directly applicable to, categorical node labels.

---

### Author Response · Authors · 2025-09-29
**Revised version has been submitted**

Dear Reviewers and Action Editor,

We sincerely appreciate the reviewers’ thoughtful comments and constructive suggestions, which have greatly improved the quality of our manuscript.
We have revised our manuscript and provided a supplementary PDF that highlights all changes.
Responses to each reviewer's comments are provided below.

Respectfully,
the authors

---

### Decision · Action_Editor_CsMV · 2025-10-16

**Recommendation:** Accept as is

**Additional Comments:**

All reviewers found the paper interesting but had a few comment about clarity and related works. The authors did a very nice reply that was appreciated by all reviewers. They agree that the paper should be published. I recommend acceptance but suggest that the authors do a check of the text for typos (for instance in Table 3 are TLE and MLE right? MLE is quite rare?, also what TLE means should be recalled in the caption of Table 3 for more self content Table).

**Audience:**

Yes

**Audience Explanation:**

This paper addresses a  problem of measuring similarity between graphs with categorical data. It is definitely of interest to the TMLR audience.

**Claims And Evidence:**

Yes

**Claims Explanation:**

The paper proposed and study a novel Wasserstein Weisfeiler-Lehman Graph Kernel for categorical data. The theoretical results have been deemed well